# Compliance of Pregnant Women with Recommendations Regarding Pertussis, Flu and COVID-19 Vaccination During Pregnancy

**DOI:** 10.3390/vaccines13050458

**Published:** 2025-04-25

**Authors:** Szymon Bednarek, Malgorzata Swiatkowska-Freund, Radosław Szelc, Patrycja Domieracka, Magdalena Tworkiewicz, Krzysztof Preis

**Affiliations:** 1Provincial Polyclinical Hospital in Toruń, Sw. Jozefa 53-59, 87-100 Torun, Poland; szymon.bednarek@wszz.troun.pl (S.B.); radoslaw.szelc@wszz.torun.pl (R.S.); patrycja.domieracka@gmail.com (P.D.); m.tworkiewicz@outlook.com (M.T.); kpreis@gumed.edu.pl (K.P.); 2The Academy of Applied Medical and Social Sciences, Lotnicza 2, 82-300 Elblag, Poland; 3Department of Gynecology, Obstetrics and Neonatology, Medical University of Gdansk, Sklodowskiej-Curie 3 a, 80-210 Gdansk, Poland

**Keywords:** vaccine, pregnancy, pertussis, influenza, COVID-19

## Abstract

Background: Many vaccines are safe and recommended during pregnancy. Transmission of maternal antibodies produced in large quantities after vaccination protects the neonate in the first months of life, until the first vaccinations in infancy. In Poland, at the time of the study, influenza, pertussis, and COVID-19 vaccines were recommended during pregnancy. Methods: The authors performed a survey study in a group of 591 post-partum women. They were asked about the safety of pertussis, influenza and COVID-19 vaccines in pregnancy. Data regarding vaccination during pregnancy according to Polish recommendations in relation to the type of vaccine were also analysed. Results: Although 50% of patients reported that, in their opinion, the pertussis vaccine is safe and recommended during pregnancy, only 17% were vaccinated. Similar results authors obtained regarding the influenza vaccine (51% and 6%, respectively). The highest knowledge and compliance with recommendations correlation to the education level was observed in women with college and university education: 65% of them thought that pertussis vaccine is safe and recommended during pregnancy; 27% of them were vaccinated; 63% of them reported that they know that influenza vaccine is recommended during pregnancy and 9% were vaccinated. In a group with the lowest education, 14% reported that the pertussis vaccine is recommended as well as 24%-influenza vaccine. No patient in this group was vaccinated during pregnancy. Conclusions: Low compliance of pregnant women was confirmed in our study, and the desperate necessity of patients’ education regarding the safety of recommended vaccines is warranted. We showed that even patients who know that the vaccine is recommended do not receive vaccination, and the lower the education level, the lower the compliance.

## 1. Introduction

The main role of vaccination is to stimulate the body’s immune response against pathogenic microorganisms. According to the most up-to-date definition of immunological terms by the CDC (US Centers for Disease Control and Prevention), vaccination is the introduction of a vaccine into the body in order to produce defence mechanisms against the given disease. In response to the vaccination, the patient produces immune antibodies, whose level usually rises for 2–4 weeks and then drops, achieving a constant low level after a few more weeks [1].

We now have vaccines against both viral and bacterial diseases. These can be vaccines containing live microorganisms or inactivated, mono- or multivalent vaccines.

Live vaccines contain attenuated viruses and bacteria that do not cause disease in immunocompetent individuals. Live vaccines are contraindicated in pregnancy as the microorganisms delivered through the vaccine can cross the placenta and cause foetal complications. Pregnancy should be avoided for one month after vaccination with a live vaccine, whereas attenuated vaccines are not contraindicated during breastfeeding [2,3].

Inactivated vaccines contain immunogenic components of an inactivated pathogen or toxoid. Inactivated vaccines are less immunogenic than live vaccines, and they are safe for pregnant women and foetuses. This group includes influenza and pertussis vaccines. However, in the first trimester of pregnancy, only absolutely necessary vaccinations should be recommended, even if they are inactivated vaccines, in order to avoid linking the vaccinations given to pregnant women to spontaneous miscarriages, which are relatively common during this period [2].

During pregnancy, complex adaptive changes necessary for immunological tolerance of the foetus by the mother’s immune system reduce the reactivity of the immune system not only to the foetus, but also to other foreign antigens [4].

Due to the impaired immune response and the characteristic changes in the cardiorespiratory system of pregnant women, pneumonia occurring in the course of influenza virus infection is much more common in pregnant women than in the general population [5,6]. The risk of hospitalisation due to influenza at 14 to 20 weeks’ gestation is 1.44 times higher, and at 37 and 42 weeks’ gestation, even 4.67 times higher than in the general population [7]. Pregnant women requiring hospitalisation for influenza have 3.82 times higher preterm birth rates. The incidence of needing a caesarean section increases by 3.47 times, while the risk of stillbirth grows by 2.5 times [8].

A study by Omer et al. showed that the number of preterm births among pregnant women vaccinated against influenza decreased significantly (adjusted Odds Ratio 0.60) [4]. According to another study, influenza vaccination during pregnancy significantly reduces the risk of prematurity, low birth weight, and the need for hospitalisation among infants up to 6 months by 81% [9,10].

The placenta is an immunologically active organ. Active transplacental transport of IgG antibodies begins at week 13 of pregnancy and increases during the pregnancy. It is a mechanism of passive foetal immunisation that protects newborns and infants from severe infections. The foetus receives most antibodies in the last four weeks of pregnancy, which justifies recommending vaccination aimed at giving immunity to newborns in the third trimester of pregnancy in order to correlate the highest levels of antibodies after vaccination with the greatest placental transfer of these antibodies. This rule cannot be applied to seasonal influenza vaccination recommendations [11,12,13,14].

A level of antibodies sufficient to protect the newborn can be successfully obtained when a pregnant woman is vaccinated. The optimum vaccination time, when the time remaining until delivery allows the patient and the foetus to obtain the maximum level of antibodies, is at the beginning of the third trimester of pregnancy [15,16]. Vaccination against pertussis at 28–32 weeks of gestation provides protection against pertussis in infants up to the first three months of life in 91% of cases, and a similar success rate (91%) was shown by the vaccine in terms of hospitalisation due to pertussis in infants up to the end of 2 months of age [17]. Immunisation of pregnant or pre-conception women with a minimum of two doses of tetanus toxoid reduces neonatal tetanus mortality by 94% [18].

Due to the increased risk of developing a severe case of certain conditions in pregnancy and the need for passive immunisation of foetuses, in many countries it is recommended to vaccinate pregnant women. In Poland, the currently (2024) recommended vaccinations during pregnancy are: influenza vaccine (single dose administered seasonally, from September to March), COVID-19 vaccine administered 12 months after the previous dose, combined Diphtheria, Tetanus, acellular Pertussis (DTaP) vaccine administered between week 27 and 36 of pregnancy and Respiratory Syncytial Virus (RSV) vaccine between week 24 and 36 [19,20]. Previously, since 2016, vaccination against influenza and diphtheria, tetanus and pertussis was recommended. COVID-19 vaccination has been recommended since 30 September 2022, with two doses given 6 months apart and booster doses once a year, regardless of whether the woman is pregnant [13]. The recommendation to vaccinate pregnant women against RSV was released in 2024.

Gram-negative pertussis vaccines are only available as combination vaccines together with tetanus toxoid and reduced diphtheria toxoid as DTaP vaccine or additionally with the polio vaccine component as DTaP-IPV (Inactivated Polio Vaccine). These vaccines are safe in pregnancy. In terms of influenza vaccines, inactivated vaccines are safe; it is not recommended to give live vaccines administered intranasally [13].

Pertussis vaccination should be given regardless of the time of the patient’s previous immunisation with this vaccine. No increased incidence of adverse reactions was observed when the pertussis vaccine was repeated after a break of at least 4 weeks from the previous DTaP vaccination [21,22]. Concurrent administration of influenza vaccine did not increase the incidence of adverse effects [23].

Vaccination of those close to a person at risk of severe infection, in this case a newborn, is referred to as cocooning [2].

The aim of this study was to assess the proportion of women vaccinated with recommended vaccines during pregnancy in the context of their knowledge of the recommended vaccines in pregnancy and their safety.

## 2. Materials and Methods

The researchers prepared a self-administered questionnaire consisting of 9 questions. The first five questions concerned details about the respondent, and the following four questions concerned the aspect under study. The study was approved by the Bioethics Committee of the Academy of Applied Medical and Social Sciences in Elbląg—Resolution of 8 January 2024 issued on the basis of application number KB4/2023. The questionnaire was anonymous and was distributed at the Obstetrics Ward of the Department of Obstetrics, Feminine Diseases and Oncological Gynaecology of The Ludwik Rydygier Provincial Polyclinical Hospital in Torun in 2024, where women after birth were hospitalised. All patients received the questionnaire, and the completed questionnaires were left by the women who wanted to participate in the survey at the ward with the doctors or midwives. Patients answered questions regarding their knowledge of the recommendations for vaccinations during pregnancy and indicated whether they had been vaccinated during pregnancy. All collected surveys were included in the study, no exclusion criteria were applied.

The responses obtained were collected in a Microsoft Excel 97-2003 spreadsheet in which data analysis was also carried out. All data were parametric and were compared using the Chi-square test.

## 3. Results

The authors analysed questionnaires completed by 591 patients. The patients were of an average age of 30.8 years (ranging from 16 to 44). Characteristics of the patients are presented in Table 1.

In the data analysis, it is noteworthy that there are more women with higher education qualifications among those giving birth to their first child than among those giving birth for the second or another time; however, the difference turned out to be statistically insignificant (56.1% of first-time mothers and 50.1% of women giving birth to at least their second child had higher education qualifications; *p* = 0.15).

Half of the respondents knew that pertussis vaccination is recommended for pregnant women (297 women, 50.2%), while 10 patients (1.7%) indicated that this vaccination was not recommended or even that it was contraindicated. Only 98 women, or 16.6% of the study group, and only 33.0% of the patients who said this vaccination was recommended in pregnancy, received vaccination against pertussis.

The distribution of the responses and vaccination varied depending on education, number of children and place of residence and is presented in Table 2. Women with higher education were statistically significantly more likely to indicate that pertussis vaccination was recommended during pregnancy and to vaccinate than women with secondary education or lower qualifications. Primiparous women, regardless of similar knowledge level to the multiparous women, were more likely to vaccinate during pregnancy.

Similar results were obtained for the influenza vaccination. In the entire study group, 50.9% of patients (301 women) said that the vaccination was recommended in pregnancy and 3.5% (21) said that it was not recommended or was contraindicated. Similar data were obtained for the influenza vaccination. In the entire study group, only 33 pregnant women were vaccinated, representing 5.6% of the respondents, and 11.0% stated that this vaccination is recommended in pregnancy. The distribution of responses and vaccination according to education level, parity and residence is presented in Table 3. More of higher educated women knew that influenza vaccination is recommended and safe in pregnancy when compared to women with lower qualifications and were more likely to vaccinate. Parity and residence did not influence responses.

COVID-19 vaccination prior to pregnancy was declared by 62.6% of the respondents (370 women). It was 70.0% among women with higher education, 55.7% among women with secondary education and 50.8% among patients with lower education. These differences were statistically significant (*p* < 0.001). There was no correlation between immunisation with the COVID-19 vaccine and the number of births (63.1% among women giving birth for the first time and 62.2% among those giving birth at least a second time) (*p* = 0.98). Urban residents were more likely to be vaccinated than rural residents at 64.8% and 58.9%, respectively, but the difference was not statistically significant (*p* = 0.35), whereas only 5 patients had been vaccinated against COVID-19 during pregnancy, which represented 0.8% of the study group and was too small a number for any statistical compilation.

## 4. Discussion

The issue of vaccinations in pregnancy is a relatively new topic, and recommendations for vaccination of pregnant women are still difficult to accept by pregnant women. Many doctors are not convinced to promote them either. Pregnant women should find out about any vaccinations recommended during pregnancy from their attending doctor, i.e., usually a gynaecologist in Poland. The level of knowledge of recommended vaccinations in pregnancy among the patients surveyed clearly indicates that doctors are not performing this task to a satisfactory degree that would lead to high vaccination rates. Only half of the respondents knew that influenza and pertussis vaccinations are recommended during pregnancy. This can be linked to the fact that, until recently, gynaecologists were not involved in the implementation of the vaccination scheme, as it was customarily the task of paediatricians and family medicine practitioners. In Canada, Baysac observed a higher vaccination rate of pregnant women whose pregnancies were managed by family physicians than those managed by gynaecologists, which supports this thesis [24].

It is surprising that, despite knowing that vaccination was recommended, 67% of women did not receive vaccination. This is certainly influenced by the trend observed in many societies to question vaccination as a mechanism for producing collective immunity at the expense of possible post-vaccination complications in the individual [25,26]. However, it does not seem that the percentage of people with negative attitudes towards vaccinations is high enough to justify the very low vaccination rates among pregnant women. The results obtained are more likely connected with the low acceptance of the concept of vaccinating pregnant women among doctors, the fact that vaccinations for pregnant women at the time of the survey were paid (unreimbursed) vaccinations, and the fact that pregnant women often had to travel between the gynaecology practice managing their pregnancy to the nearest vaccination centre, which for some pregnant women living in rural areas meant travelling to locations situated many kilometres away. However, the obstructed access to vaccination centres was not reflected in lower vaccination rates among pregnant women living in rural areas.

Razzaghi presented the vaccination rates of pregnant women against pertussis and influenza during the flu season. It is difficult to compare influenza vaccination rates with the results obtained, as the present study did not analyse the dependence of vaccination rates on the season, whereas, pertussis vaccination, which is recommended regardless of the season, was received by 56.6% of patients in the United States, which is significantly more than the under 17% recorded in the Polish population [27]. Very high vaccination rates are reported in the Danish literature, with 95% of pregnant women vaccinated against tetanus and 58% against COVID-19 [28]. In Poland, vaccination rates against COVID-19 were analysed, with 74% of patients vaccinated before or during pregnancy [29]; in the group analysed, it was lower, reaching only 62.6%.

Significantly higher vaccination rates were also reported by Baysac et al. in the Canadian population, where the percentage of patients who were informed of the pertussis vaccination recommendation and who received the vaccination was approximately 79% (compared to 33% in the Polish population) [24]. A Turkish report on influenza vaccination rates among pregnant women who have undergone vaccination consultations shows that only 54% of women who were informed of the recommendation were later vaccinated [30].

As in the material analysed, in Canada, a lower percentage of vaccinated women was found among pregnant women who had previously given birth. The dependence of vaccination rates on income reported by Baysac and Razzaghi can be compared with the dependence on education level, as in Poland, people with lower qualifications are relatively more likely to represent the lower income group. These groups in the studies analysing vaccination rates in Canada, the US, the Netherlands, Turkey, in the Nowacka study and the population analysed here were less likely to receive vaccination than those with higher economic status or higher education qualifications [24,27,28,29,30]. A higher level of education, determining a higher level of knowledge about vaccination, probably obtained not only from the doctor but also from other publicly available sources such as the Internet, is associated in the results with higher vaccination rates.

The results show the need for very intensive education of the whole population, drawing attention in vaccination promotion campaigns to vaccinations given in pregnancy and their safety. Greater awareness that vaccination is recommended and safe in pregnancy is associated with higher vaccination rates. Therefore, educational campaigns should pay attention to patient groups with lower awareness of vaccination and lower vaccination rates, i.e., patients with lower education.

Irrespective of campaigns directed at patients, more effective action should be taken to activate medical practitioners attending pregnant women in the context of vaccination promotion. Every pregnant woman should be informed of the recommendations for vaccination during pregnancy and the possibility of receiving these vaccinations at the insurer’s expense. The introduction of free pertussis vaccinations for pregnant women in Poland should increase patient interest in this vaccination and, incidentally, in other vaccinations recommended in pregnancy.

The lower vaccination rates among pregnant patients managed by gynaecologists compared to pregnant patients managed by family doctors, observed by Baysac, can be justified by the fact that family doctors also vaccinate children, are much more familiar than gynaecologists with the subject of vaccinations, and they are certainly also more confident in vaccinating pregnant women. A significant impact of interventions involving rewards for doctors who vaccinate the highest percentage of pregnant women, financial gratification, staff education, etc., on vaccination rates was observed by Mazzoni [31]. Much of the literature now focuses on efforts to combat anti-vaccine movements, assisting doctors to promote vaccination not only among pregnant women [24,32,33]. These methods are worth recommending, especially in areas with very low vaccination rates among pregnant women, whereas, in Poland, gynaecologists are mainly responsible for vaccinating pregnant women.

Our study was limited by the lack of possibility to verify the credibility of the survey responses. In Poland, there is currently a digital vaccination registry system; however, its operation during the study period was limited, and only patients and vaccination facilities had access to the database.

Vaccination rates among pregnant women in Poland are very low, which is probably due to the low involvement of doctors in informing patients about the safety of vaccination and the recommendations concerning it, and consequently, the low level of knowledge among patients about the safety of vaccination during pregnancy. In order to increase the vaccination rates among pregnant women, doctors should be motivated to inform patients about recommended vaccinations during pregnancy, aim to make it possible to vaccinate pregnant women in every gynaecological practice managing pregnancies and try to use interventions tried and tested in other countries in order to significantly increase the vaccination rates among patients.

## 5. Conclusions

Knowledge of recommended vaccinations in pregnancy among pregnant women is insufficient; therefore, it is necessary to carry out extensive educational campaigns both for patients and for doctors who subsequently provide information to pregnant women.

## Figures and Tables

**Table 1 vaccines-13-00458-t001:** Sociodemographic characteristics of the study group (N = 591).

Variable	*n* (%)
**Level of education**	
- Primary	59 (10.0%)
- Secondary	221 (37.4%)
- Higher	311 (52.6%)
**Medical background**	
- Medical	16 (2.7%)
- Paramedical	71 (12.0%)
**Place of residence**	
- Rural	224 (37.9%)
- Small town	67 (11.3%)
- Large city	300 (50.8%)
**Parity**	
- Primiparous	244 (41.3%)
- Multiparous	347 (58.7%)

**Table 2 vaccines-13-00458-t002:** Knowledge about pertussis vaccination during pregnancy and compliance with recommendations according to education level, parity and residence.

Characteristics of Respondents		Know That It Is Recommended		Vaccinated	
Education level	Higher	205 (66%)	*p* < 0.001	83 (27%)	*p* < 0.001
	Secondary	84 (38%)		15 (7%)	
	Primary	8 (14%)		0 (0%)	
Parity	Primiparous	128 (52%)	*p* = 0.67	53 (22%)	*p* = 0.02
	Multiparous	169 (49%)		45 (13%)	
Residence	Urban	191 (52%)	*p* = 0.54	65 (18%)	*p* = 0.64
	Rural	106 (47%)		33 (15%)	

**Table 3 vaccines-13-00458-t003:** Knowledge about influenza vaccination during pregnancy and compliance with recommendations according to education level, parity and residence.

Characteristics of Respondents		Know That It Is Recommended		Vaccinated	
Education level	Higher	196 (63%)	*p* < 0.001	30 (10%)	*p* < 0.001
	Secondary	91 (41%)		3 (1%)	
	Primary	14 (24%)		0 (0%)	
Parity	Primiparous	125 (51%)	*p* = 0.99	17 (7%)	*p* = 0.47
	Multiparous	176 (51%)		16 (5%)	
Residence	Urban	191 (52%)	*p* = 0.78	20 (5%)	*p* = 0.98
	Rural	110 (49%)		13 (6%)	

## Data Availability

Data available on request due to privacy restrictions.

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
