# Peer review of "Compliance of Pregnant Women with Recommendations Regarding Pertussis, Flu and COVID-19 Vaccination During Pregnancy"

_vaccines, 2025, doi:10.3390/vaccines13050458_

Round 1

Reviewer 1 Report

Comments and Suggestions for Authors

The research topic is current and interesting. The aim of this study was to assess the proportion of women vaccinated with recommended vaccines during pregnancy in the context of their knowledge of the recommended vaccines in pregnancy and their safety.

Major issues:

  1. Material and Method section: Describe how the participants were selected for the study (one by one, or other). What was the sample size calculated on? What were the inclusion and exclusion criteria for the study? What was the response rate? Supplement the statistical analysis with one of the regression analysis methods.
  2. State the limitations of the study, in the Discussion section, before the conclusion.
  3. The introduction is too long. Shorten and highlight only the key facts.

Author Response

Dear Reviewer,
We would like to thank you for your thorough analysis of our manuscript and your valuable suggestions. We have addressed your comments within the text and language corrections will be performed after acceptance of all corrections.

Comment 1: Material and Method section: Describe how the participants were selected for the study (one by one, or other). What was the sample size calculated on? What were the inclusion and exclusion criteria for the study? What was the response rate? Supplement the statistical analysis with one of the regression analysis methods.

Response 1: The study included postpartum women who gave birth in the hospital, all of them got the questionnaire.  Inclusion criteria comprised having delivered a baby and completing the survey. No participants were excluded, as all willing individuals completed the questionnaire in full. Initially, the target sample size was set at 1500 participants; however, during the course of the study, the national reimbursement policy for vaccinations during pregnancy changed in Poland, expanding access to free pertussis vaccination. This policy shift could potentially affect the validity of the statistical analysis (as the initial aim was to analyze non-reimbursed, recommended vaccinations). Therefore, the final study group comprised 591 patients.

Comment 2: State the limitations of the study, in the Discussion section, before the conclusion.

Response 2: A limitation of the study is the inability to verify the reliability of the participants’ responses. Although a national digital vaccination registry is currently operational in Poland, during the study period, access to the database was restricted to patients and vaccination centers only. We have mentioned this in lines 577–580.

Comment 3: The introduction is too long. Shorten and highlight only the key facts.

Response 3: We have reviewed and revised the introduction, shortening it and focusing solely on the most essential information and content directly related to the subject matter of the study.

Once again, we sincerely thank you for your insightful suggestions. We believe that the current version of the manuscript has improved in both clarity and value, and is now suitable for subsequent stages of the publication process. We also confirm that, following the acceptance of the revisions, the manuscript will undergo a final linguistic and grammatical revision.

Reviewer 2 Report

Comments and Suggestions for Authors

This study refers the results of a survey about vaccination among pregnant women in Poland.

The results are of interest as they show the low degree of vaccination, even though that women know the importance of vaccination. It also show that gynecologist should be more active in supporting and promoting vaccination.

There are some question to review:

  • Even though the introduction gives important information, I consider that could be reduced.
  • I deduced from the reading that the survey was done after the delivery. I think that the authors should include in the methods the moment when the survey was responded.
  • The organization of the results is somewhat a confusing. I consider that it could be better if the authors organize it by each type of vaccine. The inclusion of some tables with the main results could be also helpful.
  • Figure 1 is not referenced in the text. In fact, I consider that it is not necessary
  • Conclusion: The conclusions are in the line of the results, but could be explained in less words.
Comments on the Quality of English Language

English Language needs a review.

Author Response

Dear Reviewer,
We would like to express our gratitude for reviewing our results, providing a thorough analysis, and offering valuable suggestions. We have followed your recommendations accordingly.

Comment 1: Even though the introduction gives important information, I consider that it could be reduced.

Response 1: We have re-evaluated and shortened the introduction, focusing on the most relevant content, particularly that which is directly related to the subject of the study.

Comment 2: I deduced from the reading that the survey was done after the delivery. I think that the authors should include in the methods the moment when the survey was responded.

Response 2: We clarified the timing of the questionnaire administration and specified the exclusion criteria in lines 226–231. We hope this information now presents itself more clearly.

Comment 3: The organization of the results is somewhat confusing. I consider that it could be better if the authors organize it by each type of vaccine. The inclusion of some tables with the main results could be also helpful.

Response 3: We are aware that the amount of data presented in the manuscript may have caused some confusion. We organized it by the type of vaccination and presented some data in tables. We hope it is more clear.

Comment 4: Figure 1 is not referenced in the text. In fact, I consider that it is not necessary.

Response 4: We removed the figure

Comment 5: Conclusion: The conclusions are in the line of the results, but could be explained in fewer words.

Response 5: We have revised and streamlined the conclusion, emphasizing the most important finding that emerges from our study.

Once again, we thank you for your insightful feedback. We trust that the revised manuscript is now more coherent and valuable, and suitable for the next stage of the publication process. We also confirm that, upon acceptance of the revisions, the manuscript will undergo an additional linguistic and grammatical review.

Reviewer 3 Report

Comments and Suggestions for Authors

The manuscript requires a thorough revision of its English grammar. Numerous grammatical errors hinder readability and comprehension. A professional language review is recommended to enhance clarity and coherence.

The study does not present particularly novel findings. While it may hold some local relevance, it lacks the broader significance required for an international audience. The authors should consider emphasizing any unique contributions to justify wider interest.

The introduction is excessively long and does not effectively identify the research problem. A substantial portion is dedicated to theoretical background, which could be streamlined. The authors should focus on clearly articulating the research gap and objectives while reducing extraneous theoretical discussions.

The methodology section requires a more detailed description. The questionnaire, consisting of only nine questions on pregnant women’s opinions, appears insufficiently robust. Additionally, it is unclear whether vaccination status was confirmed among participants. Clarifying this aspect is crucial for the study’s validity.

The presentation of results is inadequate. General data and comparisons should be structured in tables for clarity. The use of a 3D graph is not recommended for publication, and alternative visualization methods should be employed to enhance interpretability.

Comments on the Quality of English Language

A professional language review is recommended

Author Response

Dear Reviewer,
We are grateful for your review of our findings, your critical analysis, and your valuable recommendations. We have addressed your comments in the revised version of the manuscript.

Comment 1: The study does not present particularly novel findings. While it may hold some local relevance, it lacks the broader significance required for an international audience. The authors should consider emphasizing any unique contributions to justify wider interest.

Response 1: The issue of vaccination in Poland remains a subject of controversy, often situated at the intersection of scientific merit and anti-vaccination movements. In our view, governmental educational campaigns have not achieved satisfactory outcomes. Based on our observations, the implementation of our study has already contributed to increased interest in the subject among the postpartum women who completed the questionnaire, and it may influence their willingness to receive vaccinations in future pregnancies. Currently, Poland has introduced reimbursement for RSV vaccination, and we are actively collecting data related to its implementation.

Comment 2: The introduction is excessively long and does not effectively identify the research problem. A substantial portion is dedicated to theoretical background, which could be streamlined. The authors should focus on clearly articulating the research gap and objectives while reducing extraneous theoretical discussions.

Response 2: Thank you for this suggestion. In the original version, we aimed to provide an overview of the current knowledge on pregnancy-related recommended vaccinations. We have since revised the introduction, shortening it significantly and focusing on the most pertinent information related directly to the study objective.

Comment 3: The methodology section requires a more detailed description. The questionnaire, consisting of only nine questions on pregnant women’s opinions, appears insufficiently robust. Additionally, it is unclear whether vaccination status was confirmed among participants. Clarifying this aspect is crucial for the study’s validity.

Response 3: In lines 226–231, we clarified the timing of questionnaire completion and the exclusion criteria. We acknowledge that a limitation of the study lies in the inability to verify the reliability of self-reported vaccination status. Although Poland has implemented a national digital vaccination registry, it was only partially operational during the study period, and access was limited to patients and vaccination centers. We addressed this issue in lines 577–580.

Comment 4: The presentation of results is inadequate. General data and comparisons should be structured in tables for clarity. The use of a 3D graph is not recommended for publication, and alternative visualization methods should be employed to enhance interpretability.

Response 4: We recognize that the volume of data may be overwhelming in textual format. In response to your recommendation, we have reorganized the section, added tables to clearly present the most important findings of the study. We have also removed the figure.

Once again, we sincerely thank you for your constructive feedback. We are confident that the revised manuscript is now more transparent and valuable, and we believe it is suitable for progression to the next stage of the publication process. We also confirm that, following acceptance of the revisions, the manuscript will undergo a final linguistic and grammatical review.

Round 2

Reviewer 1 Report

Comments and Suggestions for Authors

I have no any other concerns.

Reviewer 2 Report

Comments and Suggestions for Authors

The authors have done important changes in the manuscript. Now is more clear and the results can be better understood.